# Rice *GA3ox1* modulates pollen starch granule accumulation and pollen wall development

**Kun-Ting Hsieh**[1,2], **Chi-Chih Wu**[2], **Shih-Jie Lee**[1], **Yu-Heng Chen**[1], **Shiau-Yu Shiue**[1], **Yi-Chun Liao**[1], **Su-Hui Liu**[1], **I.-Wen Wang**[3], **Ching-Shan Tseng**[3], **Wen-Hsiung Li**[2,4], **Chang-Sheng Wang**[5,6], **Liang-Jwu Chen**[1,6]*

1 Institute of Molecular Biology, National Chung Hsing University, Taichung, Taiwan, 2 Biodiversity Research Center, Academia Sinica, Taipei, Taiwan, 3 Division of Biotechnology, Taiwan Agriculture Research Institute, Taichung, Taiwan, 4 Department of Ecology and Evolution, University of Chicago, Chicago, Illinois, United States of America, 5 Department of Agronomy, National Chung Hsing University, Taichung, Taiwan, 6 Innovation Center for Agricultural Biotechnology, National Chung Hsing University, Taichung, Taiwan

* ljchen@nchu.edu.tw

**Data Availability Statement:** All relevant data are within the paper and its Supporting information files.

## Abstract

The rice GA biosynthetic gene *OsGA3ox1* has been proposed to regulate pollen development through the gametophytic manner, but cellular characterization of its mutant pollen is lacking. In this study, three heterozygotic biallelic variants, "-3/-19", "-3/-2" and "-3/-10", each containing one null and one 3bp-deletion allele, were obtained by the CRISPR/Cas9 technique for the functional study of *OsGA3ox1*. The three homozygotes, "-19/-19", "-2/-2" and "-10/-10", derived from heterozygotic variants, did not affect the development of most vegetative and floral organs but showed a significant reduction in seed-setting rate and in pollen viability. Anatomic characterizations of these mutated *osga3ox1* pollens revealed defects in starch granule accumulation and pollen wall development. Additional molecular characterization suggests that abnormal pollen development in the *osga3ox1* mutants might be linked to the regulation of transcription factors OsGAMYB, OsTDR and OsbHLH142 during late pollen development. In brief, the rice *GA3ox1* is a crucial gene that modulates pollen starch granule accumulation and pollen wall development at the gametophytic phase.

## Introduction

Pollen is essential for plant reproduction, and rice pollen is comprised of a multilayered pollen wall and is full of starch granules in the central cytoplasm. Pollen wall is a sturdy structure and functions to protect pollen from various environmental stresses, and the accumulated starch granule provides energy for pollen germination and pollen tube elongation. The pollen wall formation starts from an early stage of pollen development, and the development of the outer layer, exine, is mainly regulated through tapetum, a sporophytic tissue [1], while the inner layer, intine, is thought to be determined by the male gametophyte [1]. In contrast, starch is synthesized at the late stage of pollen development [2], and its accumulation is mainly regulated by male gametophyte [3]. Rice mutants with defects in pollen wall formation [1] or starch

**Funding:** This work was financially supported by grants (MOST107-2313-B-005-016-MY3 to Liang-Jwu Chen) from Ministry of Science and Technology, Taiwan and (AS-TP-109-L10 and AS-KPQ-111-ITAR-111100 to Wen-Hsiung Li) from Academia Sinica, Taiwan, and in part by the Advanced Plant Biotechnology Center via the Featured Areas Research Center Program within the framework of the Higher Education Sprout Project by the Ministry of Education (MOE), Taiwan. The funders had no role in the study design, data collection or analysis, decision to publish, or preparation of the manuscript.

**Competing interests:** The authors have declared that no competing interests exist.

synthesis [4, 5] exhibit male sterility. Moreover, the involvement of gibberellin (GA) in pollen wall formation [6, 7] and starch synthesis [8] has been reported. However, how GA affects these processes in gametophyte is not well understood.

It has been known that pollen development is regulated by intricate networks involving GA biosynthesis and GA signaling pathways [7, 9]. Many GA-deficient and GA-signaling mutants cause abnormal pollen development [9]. For example, the GA-deficient *oscps* mutant affects stamen and pollen development [7, 8], and a mutation of GA-regulated transcription factor (TF) GAMYB causes defects in tapetal programmed cell death and pollen formation in rice [7, 10]. GAMYB functions as a positive regulator of TAPETUM DEGENERATION RETARDA-TION (TDR) [7] and OsbHLH142 [11] to control pollen development [12, 13]. In addition, a reduction in GA caused by low temperature could disrupt rice pollen development, leading to male sterility and yield loss [14]. These lines of evidence suggest that the presence of GA and its functional regulation are critical for pollen development. The functions of GA were studied mainly at the early stage of pollen development [7], which affects pollen wall formation in sporophytic manner. However, little is known about the gametophytic GA functions in late pollen development.

The anther is the major floral organ for GA biosynthesis. High concentrations of bioactive GA have been detected in rice anthers [15, 16], and many GA biosynthesis genes, such as *OsCPS*, *OsKS*, *OsKO2*, *OsKAO*, *OsGA20ox1*, *OsGA20ox3* and *OsGA3ox1*, were highly expressed in anthers [16, 17], suggesting that these genes are responsible for anther GA accumulation. Among these genes, null mutants of *OsCPS*, *OsKS*, *OsKO2* and *OsKAO* exhibited a severe dwarf phenotype [18, 19]; mutation of *OsGA20ox3* showed semidwarfism with a low seed-setting rate [19], but no further characterization of their anthers has been reported. No mutant of *OsGA20ox1* is presently available. It is known that *OsGA3ox1* is expressed mainly in anthers during the late stage of pollen development, while *OsGA3ox2* is preferentially expressed in pistil and other vegetative organs and in anthers at the early stage of pollen development [14, 16, 20, 21]. The *d18* mutants of *OsGA3ox2* showed a severe dwarf phenotype [18, 20] but no anther characterization is available. A recent report showed that mutation of *OsGA3ox1* did not affect plant growth but severely affected pollen viability, pollen germination and seed fertility [21], suggesting that OsGA3ox1 is involved only in pollen development. Unlike mutations of other genes, such as *oscps*, *osgid1* and *gamyb*, which regulate pollen development during the sporophytic phase [7], the *OsGA3ox1* was proposed to regulate pollen development during the gametophytic phase [21]. However, no cellular observation on sporophytic and gametophytic features in the *osga3ox1* mutant has been made. In the present study, we created various *osga3ox1* mutants by the CRISPR/Cas9 technique, and our analyses of these mutants indicated that *OsGA3ox1* has gametophytic functions and modulates pollen starch granule accumulation and pollen wall development.

## Materials and methods

### Plant materials, growth conditions and agronomic trait investigations

The rice cultivar *Oryza sativa* Tainung 71, a japonica-type aroma rice bred by scientists at the Taiwan Agricultural Research Institute (TARI), was used as the wild type in the study. The seeds of wild-type and various *osga3ox1* mutants were germinated in a growth chamber at 30˚C with a 16 h/8 h light/dark cycle for 14 days and then transferred to a greenhouse or an isolated paddy field at TARI.

The agronomic traits, including the plant height, heading date and yield-related traits, such as panicle numbers/per plant, grain numbers/per panicle, seed-setting rate (SSR) and thousand-grain weight (TGW), were measured and collected for statistical analysis. Plant height:

The plant heights were measured at the yellow-ripening/mature stages. Grain numbers/per panicle: total number of seeds per plant divided by the panicle number/per plant to obtain the averaged grain number/per panicle. Seed-setting rate (SSR): the ratio of fertile seeds vs total seeds per plant. Thousand-grain weight (TGW): the weight of 1000 fertile seeds.

## Plasmid construction and rice transformation

The sgRNA of OsGA3ox1 was designed using E-CRISP [22] (http://www.e-crisp.org/E-CRISP/index.html), and the location and DNA sequence of the sgRNA are shown in S1A Fig. Two complementary sgDNA sequences with 5' added sequences (GGCA for the forward sequence and AAAC for the reverse sequence) (S1 Table) were synthesized, annealed by a thermal cycler and integrated into the *BsaI*-cut CRISPR/Cas9 vector pRGEB32 [23]. After ligation and *E. coli* transformation, constructs were screened by *BsaI* restriction digestion and verified by sequencing. The constructed plasmid vectors were transformed into *Agrobacterium tumefaciens* strain EHA105 for rice plant transformation as previously described [24].

## Regeneration of transgenic rice and selection of gene-edited plants

Transgenic plants containing the T-DNA vector construct were obtained through 50 mg/L hygromycin selection, and the transgene was confirmed by PCR analysis of the construct and/or hygromycin phosphotransferase (*hpt*) (S1B and S1C Fig). Regenerated T0 transgenic plants from different calli were assigned different line numbers.

For the selection of gene-edited plants, we modified the polyacrylamide gel electrophoresis-based (PAGE) method previously described [25] and used the modified method to select candidates of gene-edited plants. Briefly, the target region of sgRNA was PCR amplified, and the ideal size of the PCR product used in this method was approximately 300 to 400 bp. The PCR products from transgenic plants were subjected to 3% agarose gel electrophoresis for an hour at 100 voltages. After electrophoresis, the PCR product from transgenic plants was isolated into homoduplexes (consistent or smaller than expected size) and heteroduplexes (larger than expected size) (S1C Fig), which were used to identify insertion/deletion (IN/DEL) events from transgenic plants and infer that the target gene in analyzed plants is homozygous or heterozygous [25].

To identify frameshift mutations, PCR products with clear IN/DEL events were subjected to DNA sequencing, and the sequences in PCR products were decoded via CRISPR-ID [26] (http://crispid.gbiomed.kuleuven.be/). Once putative frameshift mutations were found, the PCR products were cloned into the pGEM®-T Easy Vector (Promega, Madison, WI, USA), and ten colonies from the respective PCR products were sent for DNA sequencing for sequence confirmation.

## Pollen viability assay

The pollen viability assay was conducted as described previously [8]. Before flowering, anthers in a spikelet were collected and pollens removed from anthers were evenly placed on a glass slide, stained with staining solution (1% (v/v) of $I_2$ in 3% (v/v) KI), and then examined using a light microscope. Twenty views (12 $mm^2$/each) for the respective samples were selected for counting. The black-stained pollen is referred to as viable pollen.

## RNA extraction and gene expression analysis

Mature rice anthers (approximately 1.5 to 2.0 mm) from wild-type and *osga3ox1* mutant plants were used for RNA extraction. Total RNA was extracted using TRIzol reagent (Invitrogen,

Carlsbad, CA, USA), and RNA samples were treated with RNase-free Dnase I (Thermo Fisher Scientific, Waltham, MA, USA) to remove possible DNA contaminants. For reverse transcription, 1 μg of total RNA was used in cDNA synthesis using a RevertAid First-Strand cDNA Synthesis Kit (Thermo Fisher Scientific, Waltham, MA, USA), and the reaction was conducted following the protocol of the manufacturer in a 20 μL reaction volume. The quantitative reverse transcription PCR (RT-qPCR) was conducted as previously described [27] using one microliter of synthesized cDNA with the respective primer sets (S1 Table). The expression levels were normalized using the threshold cycle ($C_t$) values obtained for the reference gene *OsActin*, and relative quantification was conducted by the $2^{-\Delta\Delta Ct}$ method [28].

## Semithin and ultrathin sections of rice anthers and pollen grains

Mature rice anthers (approximately 1.5 to 2.0 mm) were fixed in 1% glutaraldehyde in 0.1 M phosphate-citrate buffer, pH 7.2, in a microwave tissue processor, Pelco BioWave Pro (TED PELLA, Inc., Redding, CA, USA) at a microwave power of 250 watts for 1 minute twice. After three buffer rinses (at 250 watts for 40 seconds; hereafter the buffer wash was under this condition), the samples were postfixed in 1% OsO4 in the same buffer for 1 minute at 100 watts followed by three washes of buffer. Samples were then dehydrated in an ethanol series under the condition of the power of 100 watts for 40 seconds each step. Subsequently, samples were embedded in a series of Spurr's resin at room temperature, and polymerized blocks were sectioned with a Lecia Reichert Ultracut S or Lecia EM UC7 ultramicrotome. The semithin sections (1 micrometer) were obtained and stained with 1% toluidine blue, and images of the sections were taken by a Zeiss fluorescence microscope Z1 equipped with a Zeiss Axiocam HRc CCD digital camera (MICRO-OPTICS, Fresh Meadows, NY, USA). The ultrathin sections (70~90 nanometers) were stained with 5% uranyl acetate in 50% methanol and 0.4% lead citrate in 0.1 N NaOH. Images were taken using an FEI G2 Tecnai Spirit Twin transmission electron microscope at 80 kV equipped with a Gatan Orius CCD camera.

## Measurement of starch granules

The numerical features of starch granules and pollen grains were quantified by counting and measuring starch granules and pollen grains on histological images of semithin rice anther sections taken by a Zeiss fluorescence microscope Z1 equipped with a Zeiss Axiocam HRC CCD digital camera. Quantification of pollen grains and starch granules within pollen grains was automated using a Cellprofiler ver. 2.4.0rc1 pipeline [29]. The basic steps of the pipeline were the segmentation of objects, including pollen grains and starch granules, identification of objects, and quantification of objects. The outline of the pollen grains was extracted by using duplicated mask images of pollen grains on histological images. The outline of the starch granules on sections was extracted by enhancing brightness and contrast. The pollen grains and starch granules were then segmented using an adaptive local thresholding method based on ellipse fit [30], and identified objects were counted and measured.

## Measurement of exine and intine thickness

The exine and intine thicknesses of the pollen wall were measured using Fiji [31]. At least 5 images of pollen grain thin sections and 3 to 60 sampling sites from each image were selected for measurement. Each sampling site was evenly distributed, and the thickness of the exine was measured between two flat lines to avoid spikes (see the bracket indicated in Fig 4); the thickness of the intine was measured from the base to its peak (as indicated by the blue line in Fig 4).

## Statistical analysis

The statistical comparisons were performed using one-way ANOVA and then against the WT control by Student's *t test* and graphing through the GraphPad Prism 8 algorithm (GraphPad Software Inc. La Jolla, CA, USA). The distributions of each sample dot are shown. Values are the mean±SD. The significant differences are indicated by *P* values: $*p < 0.05$, $**p < 0.01$, $***p < 0.001$.

# Results and discussion

## Creation of *OsGA3ox1* mutants

To study the biological functions of *OsGA3ox1* in rice, the plasmid pRGEB32 containing an sgRNA, which targets the first exon of *OsGA3ox1* (S1A and S1B Fig), and the CRISPR/Cas9 construct was used to create loss-of-function transgenic plants. More than 10 transgenic rice lines (T0 plants) were successfully regenerated via Agrobacterium-mediated transformation. The transgenic rice lines showing targeted gene editing were screened through a modified gel-based screening method. Among these T0 transgenic lines, 8 independent lines with different editing types, revealed by the various PCR band patterns, were selected for further characterization (S1C Fig). Their DNA sequences at the *OsGA3ox1* gene target site were sequenced, and 7 lines of different combinations of edited sequences were identified (S1D Fig).

Except for the "-3" allelic variant (a 3-bp deletion), the amino acids derived from other edited allelic variants showed frame-shift and premature stops (S1E Fig), resulting in knocked-out function of the *OsGA3ox1* gene. Two lines (#4 and #5) showed biallelic "-1/-2" and "-1/-4" variants that had both allelic genes knocked out and produced no seeds (S1C Fig). Another two lines, #7 (-3/-16/?) and #8 (-3/-31/?), each contains the "-3" allele, a knockout allele and a not-yet-identified allelic variant could produce fertile seeds, but due to its complications in editing, no further characterization was pursued. The remaining four heterozygotic biallelic lines, line #1 ("-3/-19"), lines #2 and #3 ("-3/-2") and line #6 ("-3/-10"), which also contained the "-3" allele, could produce fertile seeds and were used for further phenotypic and molecular analyses to investigate the function of *OsGA3ox1*.

## Mutation of *OsGA3ox1* affects the seed-setting rate but not plant height

Although the "-3" allele-containing heterozygotic lines could produce T1 seeds, their seed-setting rate (SSR) was significantly reduced (S1C Fig). To confirm that the low SSR was due to the loss of function of *OsGA3ox1*, T1 progenies from line #1 ("-3/-19") were first grown in a greenhouse and characterized. The "-3/-19" variant segregated into "-3/-3" homozygous, "-19/-19" homozygous and "-3/-19" heterozygous plants that were recognized by the PCR band patterns and confirmed by sequencing (S2A Fig). Our results showed that the plant heights were not different among these three genotypes; however, their SSRs were reduced, and the "-19/-19" knockout mutant produced no fertile seeds (S2B Fig).

To further confirm these observations, the T2 progenies segregated from the biallelic "-3/-19" plants, and T1 progenies segregated from biallelic "-3/-2" (line # 3) and "-3/-10" (line # 6) lines were grown in the paddy field for characterization. Genotypes of segregated progenies were identified by sequencing (Fig 1A–1C). All different variants/genotypes showed the same plant height as WT (Fig 1D–1F), similar to the plant height observed in the greenhouse. However, all homozygotic knockout variants ("-19/-19", "-2/-2" and "-10/-10") showed unfertilized panicles and produced few fertile seeds (Fig 1G, the fertile seeds are circled). The SSRs of "-19/-19", "-2/-2" and "-10/-10" were greatly reduced to only ~2.8%, 3.4% and 2.2%, respectively (Fig 1H and 1I).

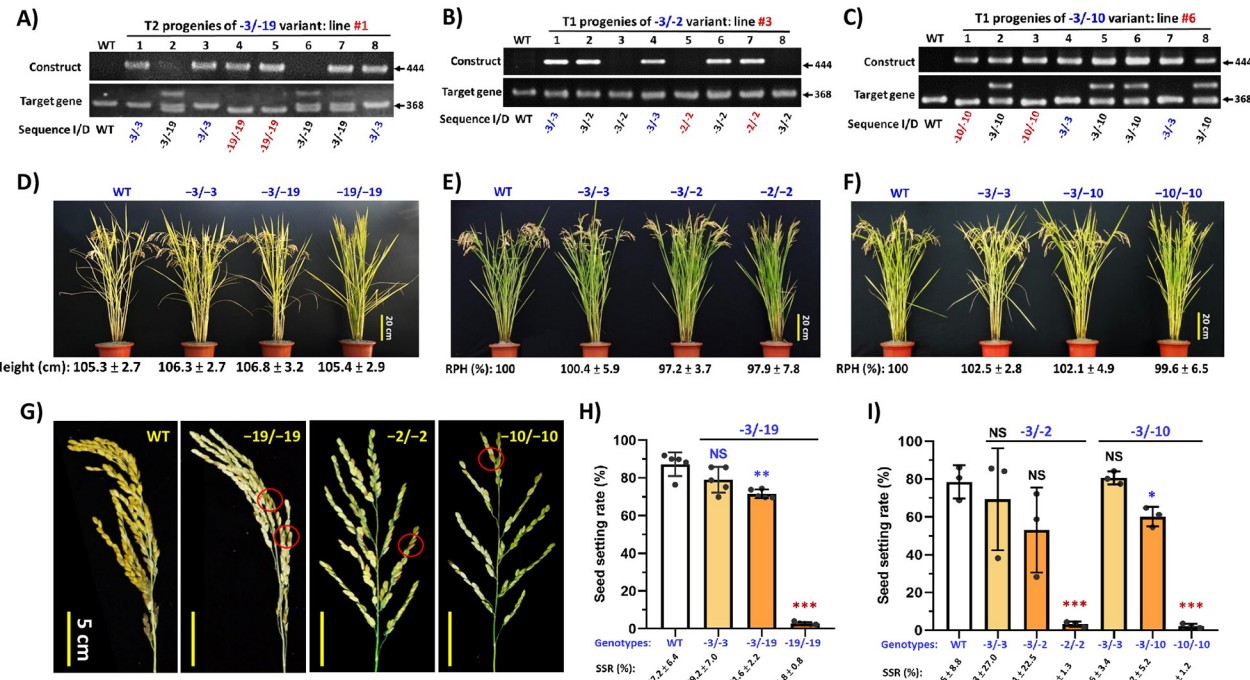

**Fig 1. Characterizations of Cas9-induced *osga3ox1* "-3/-19", "-3/-2" and "-3/-10" biallelic variants and analysis of their progenies. A) to C)** PCR product analysis of progeny plants derived from three Cas9-induced *osga3ox1* variants. Eight representative progeny plant samples from each of the three biallelic variants are shown. The progeny genotypes derived from "-3/-19" (A) and "-3/-2" (B) and "-3/-10" (C) were confirmed by sequencing and are indicated below the gel samples. The PCR primers used for construct and target gene detection are provided in S1A and S1B Fig and S1 Table. Genotypes are presented as sequence I/D (I, insertion "+" or D, deletion "-"; WT, wild-type plant or gene). Sequence variations indicated by Arabic numerals separated by a slash are used to represent the diploid allelic genotypes. **D) to F)** Representative matured plants of different genotypes segregated from three Cas9-induced *osga3ox1* variants. The knockout variants "-19/-19" (D), "-2/-2" (E) and "-10/-10" (F) showed unfertilized erected panicles. Their average plant height and relative plant height (RPH) are listed below. Sample size n≧5. G) Matured panicle comparison in WT and different knockout variants. The very few fertilized seeds in the knockout variants are circled. H) and I) Comparisons of the seed-setting rates in WT and three Cas9-induced *osga3ox1* variants. The data of H) and I) were collected from different crop seasons with different sample sizes (H, n = 5; I, n = 3). The significant differences between WT and variants were determined by Student's *t* test. *$p<0.05$, **$p<0.01$, ***$p<0.001$, NS; no significant difference.

Detailed agronomic traits of the WT host and progenies segregated from the transgene-free "-3/-19" variant/line were compared, and the plant height, panicle number and grain number/per panicle were the same for WT and all mutant genotypes (Table 1). For the SSR, "-3/-3" was similar to the WT, "-3/-19" was slightly lower than the WT, and the "-19/-19" knockout

**Table 1. Agronomic trait comparisons between TNG71 and progenies segregated from the biallelic "-3/-19" variant.**

| Host/Variants* | Genotypes** | Plant height (cm) | Panicle number-A | Grain number/per panicle-B | Seed setting rate (%)-C | 1000-grain weight (g)-D | Grain yield/per plant (g)-E |
|---|---|---|---|---|---|---|---|
| TNG71/"-3/19" Variant | WT | 105.3 ± 2.7[a] | 10.8 ± 2.2[a] | 115.2 ± 13.0[a] | 87.3 ± 6.2[a] | 24.9 ± 0.2[ab] | 27.3 ± 8.0[a] |
| | "−3/−3" | 106.3 ± 2.7[a] | 14.2 ± 5.6[a] | 107.0 ± 6.9[a] | 79.0 ± 6.8[ab] | 26.6 ± 0.8[a] | 31.5 ± 11.1[a] |
| | "−3/−19" | 106.8 ± 3.2[a] | 12.2 ± 1.9[a] | 110.2 ± 8.9[a] | 71.6 ± 2.3[b] | 26.8 ± 0.5[a] | 25.7 ± 4.1[a] |
| | "−19/−19" | 105.4 ± 2.9[a] | 13.4 ± 4.3[a] | 107.6 ± 6.5[a] | 2.8 ± 0.7[c] | 24.0 ± 1.8[b] | 20.9 ± 0.2[b] |

*Segregated progenies from the "-3/-19" variants and TNG71 (host plant) were grown in paddy fields of TARI in the second season of 2020.

**Progenies with different genotypes: homozygotes "-3/-3", heterozygotes "-3/-19" and homozygotes "-19/-19" were segregated from the Cas9-induced *OsGA3ox1* biallelic "-3/-19" variant. Grain yield E = (A x B x C/100 x D)/1000. The values are the means ± SEs (n≧5). The different superscript letters in each trait column indicate significant differences between samples according to Tukey's HSD test ($p < 0.01$).

mutant showed a significant reduction. The 1000-grain weight and grain yield per plant in the knockout mutant were significantly affected (Table 1). Taken together, our findings indicate that *OsGA3ox1* plays a critical role in the development of seeds but not in the development of vegetative organs.

### Mutation of *OsGA3ox1* affects the pollen number and pollens were stained lighter in *osga3ox1* mutants

To determine whether the low SSR was due to defects in any gametophytic features, the flower organs of the three genotypes segregated from the progeny of the transgene-free "-3/-19" variant were examined. Unlike the abnormally developed floral organs observed in GA-deficient mutants reported previously [9, 32, 33], the anthers and pistils in the three genotypes showed no difference from WT (Fig 2A). However, the total pollen and the pollen viability by IKI staining in the knockout mutant "-19/-19" were significantly reduced (Fig 2B). The numerical quantification data of the collected samples showed that the pollen numbers and the pollen viability of "-3/-19" and "-19/-19" were approximately 72.8% vs 42.6%, and 79.1% vs 51.3%, respectively, compared to the WT (Fig 2C and 2D), whereas no significant difference between

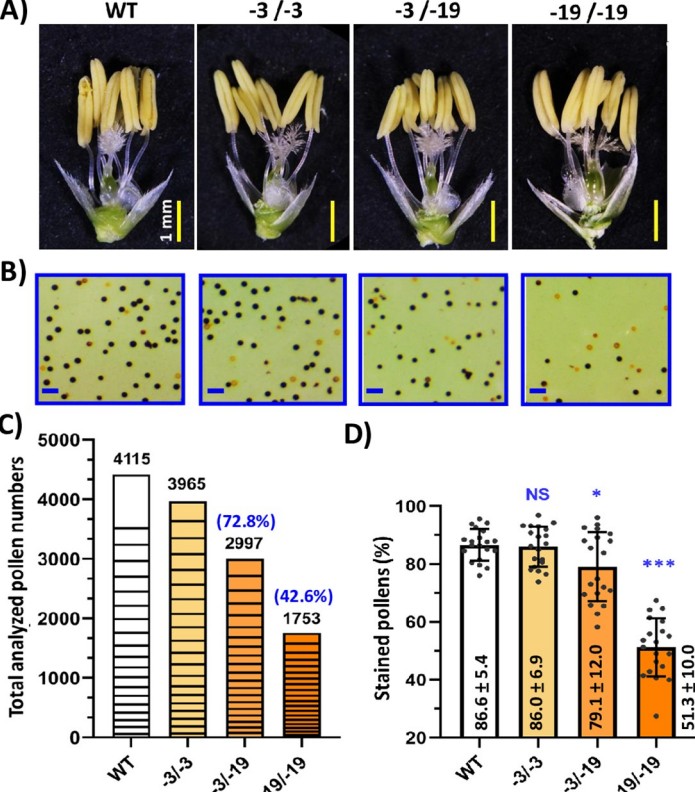

**Fig 2. Phenotypes of floral organs and characterization of pollen viability and pollen number analysis in the "-3/-19" variant. A)** Comparisons of floral organs. Flower samples from WT and different genotypes segregated from the "-3/-19" biallelic variant are shown. No phenotypic difference was observed. Bar = 1 mm. **B)** Comparisons of pollen viability. Representative views (2 mm$^2$) for each genotype are shown. The pollen viability was based on KI starch staining. Healthy pollen is stained black. Bar = 150 μm. **C)** Comparisons of total analyzed pollen numbers. Twenty views (12 mm$^2$/each) of pollen slides from different genotypes were sampled. The graphic bar of each genotype shows the total analyzed pollen numbers that were accumulated by each of their 20 samples. **D)** Comparisons of the pollen viability/stained pollens in WT and mutant genotypes. The significant differences between WT and variants were determined by Student's *t test*. *$p<0.05$, **$p<0.01$, ***$p<0.001$; NS, no significant difference. (n = 20).

"-3/-3" and WT was observed (Fig 2C and 2D). These observations suggest that *OsGA3ox1* affects pollen development and reduces SSR, but does not have any discernable effect on other floral organs.

While the SSR of the knockout mutants were dramatically decreased (Fig 1H), a high proportion (~51.3%) of the stained pollen were observed in the present study (Fig 2D) which differs from the low proportion (~5%) reported previously [21]. This variation could be due to the different rice cultivars (TNG71 vs Nipponbare) and culture environments (Taichung Taiwan 24°0′N 120°7′E vs Nagoya Japan 35°2′N 136°9′E). Furthermore, besides a few heavy stained pollen, most pollens in the knockout mutant were stained lighter than WT (Fig 2B) and these pollens were counted as stained pollen even though they might not be viable. Additionally, the total pollen numbers in *osga3ox1* mutant were dramatically decreased to ~42.6% of the WT (Fig 2C), and together with these less viable pollens, low SSR was observed in our knockout mutants.

Unlike both of the GA deficiency mutant *rpe1* and GA signaling mutant *Slr1-d3* that impaired the pollen viability but not the number of pollen grains [8], the knockout mutant *osga3ox1* reduces the total number of pollen to ~42.6% of the WT (Fig 2B and 2C). Although *OsGA3ox1* is mainly expressed in anthers at the late stage of pollen development, its expression could also be detected in the tapetum at the early stage of pollen development [16, 20]. The expression of *OsGA3ox1* has been observed in anthers from the meiosis to the tricellular stages, increasing gradually and peaking at the binucleate stage, then declining at the tricellular stage [14]. In the present study, the tapetum layer in *osga3ox1* and WT were all degenerated in the mature anther (Fig 3A), but we could not rule out the possibility that the bioactive GAs required for early pollen development could partly contributed by *OsGA3ox1*. Therefore, the bioactive GAs in *osga3ox1* mutant might be reduced to retard the progression of the tapetum program cell death (PCD), causing less nutrients for early pollen development and resulting in early pollen degeneration, which leads to pollen number reduction in *osga3ox1* mutant (Fig 2C).

When analyzing progeny segregation of the heterozygous variants, we always observed a very low ratio of the gene knockout plants in the populations. An example of 24 progenies with a segregation ratio of 11:11:2 = "-3/-3":"-3/-19":"-19/-19" was observed from the progeny of the "-3/-19" heterozygous variant (S3 Fig), which did not follow the 1:2:1 Mendelian segregation ratio. This low transmission frequency of the *osga3ox1* knockout plants in their progeny population could be due to its low pollen viability. It is similar to the observation of GA-deficient mutants, such as *cps*, *ks*, *ko2 and kao* of GA biosynthesis genes, and their mutation effects are transmitted to progeny in a gametophytic manner that affects their transmission frequency [8].

## Mutation of *OsGA3ox1* affects pollen starch granule accumulations

To pursue the cause of low pollen viability in *osga3ox1* mutant, we investigated the anatomic features of mature anthers from different genotypes. We found no difference in sporophytic features, including epidermal, endothecium and tapetum cells of anthers (Fig 3A and S4B Fig). Interestingly, there was a significant difference in the gametophytic development of pollen (Fig 3A). In contrast to WT and "-3/-3" variants, which had larger pollens and a great amount of starch granule accumulation, the pollens from the homozygotic knockout variant "-19/-19" were smaller and had less starch granule accumulation (Fig 3A and 3B), and pollens which contain small amounts of starch granules were lightly stained (Fig 2B). The quantification data showed that the pollen collected from the knockout variant had its size reduced to ~ 70% - 80% of the size of the WT (Fig 3C).

Another observation was that both the "-3" and "-19" types of pollen can be observed in the "-3/-19" variant (Fig 3A). The area ratio of starch granules to a pollen grain in different

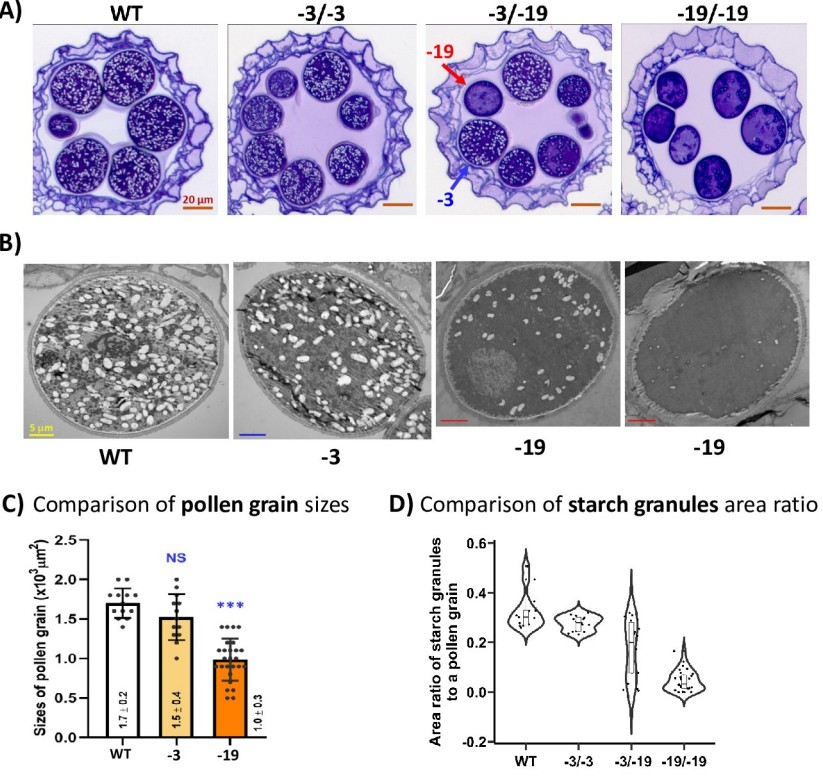

**Fig 3. The pollen grain phenotypes, sizes and starch granule distributions in "-3/-19" variants.** A) Phenotypical pollen grain comparisons in anther microscopic sections. Anther samples were from WT, and different genotypes segregated from the "-3/-19" heterozygous/biallelic variant. Two types of pollen grains, "-3"-like (blue arrow) and "-19"-like (red arrow), observed in anthers of the "-3/-19" genotype are indicated. Bar = 20 μm. B) TEM sections of pollen grains from WT, "-3" and "-19". Different types of pollen grains from "-19" are presented to show various amounts of starch granules. Bar = 5 μm. C) Comparisons of pollen grain sizes. Sizes were measured on the surface areas of the largest pollen section images available from each genotype. Sample sizes were n = 13, 13 and 28 for WT, "-3" and "-19", respectively. The significant differences between WT and mutant genotypes were determined by Student's *t test*. *$p < 0.05$, **$p < 0.01$, ***$p < 0.001$; NS, no significant difference. D) Quantification of the area ratio of starch grains to pollen grains from anther samples of WT and different genotypes. Sample sizes were n = 13, 16, 18 and 28 for WT, "-3/-3", "-3/-19" and "-19/-19", respectively. The sample dot distribution is shown in the plot.

genotypes was measured to indicate the amount of starch granule accumulation. Relatively high levels of starch accumulation (area ratio of 0.3 and above) were observed in the pollen of WT and "-3/-3", while the "-19/-19" genotype revealed an area ratio of less than 0.1. The starch accumulation ratio in pollen collected from the "-3/-19" genotype (Fig 3D) had a distribution from 0 to ~ 0.5 and formed two subgroups, in which the higher subgroup was equivalent to the group of WT and "-3/-3" and the lower subgroup was similar to the group in the "-19/-19" genotype (Fig 3D), indicating that both normal and abnormal pollen could be generated in the same anther sac. The above phenotypic observations in the "-3/-19" variant were also found in the "-3/-10" and "-2/-2" variants (S4 Fig). The pollens from "-2/-2" variant contain less starch granule and are smaller, the same as those from "-19/-19". The pollens produced from "-3/-10" variant contain two types of pollens, the same as those generated from "-3/-19" variant (S4 Fig). These data support that the defect in pollen development was indeed caused by the defect of *OsGA3ox1*.

Unlike pollen development in the GA-deficient mutant *oscps1-1*, which showed an enlarged tapetum layer and collapsed microspores [7], no morphological difference in the tapetum layer was observed in *osga3ox1* (Fig 3A). Our findings suggest that mutation of *OsGA3ox1* does not

inhibit the tapetum PCD, but is involved in the regulation of starch synthesis during late pollen development. This notion is supported by the fact that *OsGA3ox1* is expressed mainly in the late stage of pollen development [14, 16, 20] and the accumulation of pollen starch granules can be enhanced in pollens of the GA₃-treated almond anther [34]. Therefore, *OsGA3ox1* is suggested to be expressed in the pollen grains and is involved in late pollen development, mainly affecting starch synthesis (Fig 3).

## *OsGA3ox1* affects pollen wall development and starch synthesis

The rice pollen wall is elaborated and contains different substructures, including exine (outer layer), tryphine (in the cavities of exine) and intine (inner layer). Exine is further divided into two components, sexine (tectum and baculum) and nexine. Since GA has been proposed to regulate exine formation and GA-deficient mutants showed defective pollen cell walls [7], we compared the pollen wall ultrastructure between mutant *osga3ox1* and WT. From TEM image observations, the knockout mutant "-19/-19" revealed a thinner exine, but the thickness of the intine was similar to those of "-3/-3" and WT (Fig 4A). The quantification data measured from 30 to 186 samples showed that the exine thickness in the knockout mutant was ~ 81 to 89% of that in the WT, the reduction being statistically significant (Fig 4B), but no significant difference was observed in the thickness of the intine (Fig 4C).

In addition to the thinner exines, the bacula in the knockout mutant is not well organized, showing either fewer or incompletely developed bacula with abnormally connected tryphines

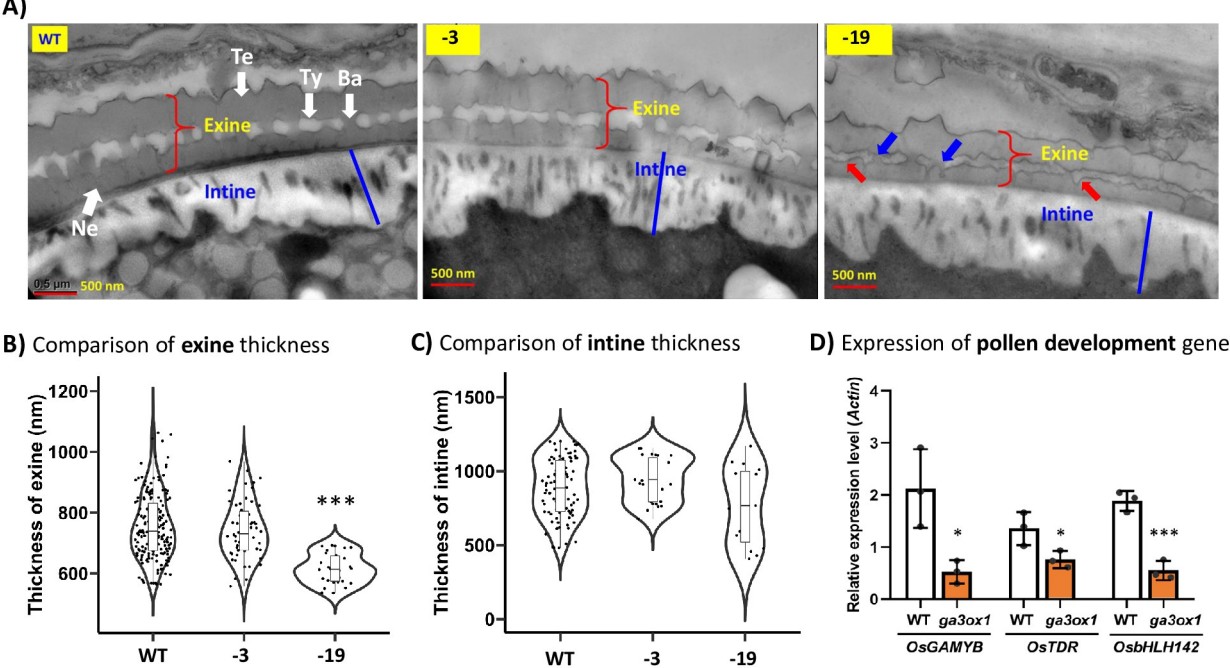

**Fig 4. The ultrastructure of pollen walls and expression analysis of pollen development genes in *osga3ox1* mutants.** A) The ultrastructure, including the exine and intine, of the pollen walls from WT, "-3" and "-19" are indicated. The structure of exine including tectum (Te), tryphine (Ty), baculum (Ba) and nexine (Ne) are indicated in WT. The abnormal formation of baculum (blue arrow) and tryphine (red arrow) in "-19" pollen are pointed. Bar = 500 nm. B) Comparison of exine thickness in WT, "-3" and "-19". Sample sizes were n = 186, 53 and 30 for WT, -3 and -19, respectively; sample dot distribution is shown in the plot. C) Comparison of intine thickness in WT, "-3" and "-19". Sample sizes were n = 90, 27 and 15 for WT, -3 and -19, respectively; sample dot distribution is shown in the plot. D) Expression analysis of the pollen development genes in WT and the *ga3ox1* knockout mutant. The significant differences between WT and variants were determined by Student's *t* test. *$p<0.05$, **$p<0.01$, ***$p<0.001$.

(Fig 4A and S5 Fig, blue arrows). While the tryphines in the WT and "-3" pollen walls are well arranged with smooth boundaries, the tryphines in the mutant show distorted and wiggly structures (Fig 4A and S5 Fig, red arrows). The ultrastructure of the "-19"-like pollen from "-3/-19" anther was phenotypically similar to that of the "-19" pollen from "-19/-19" anther (S5 Fig).

The exine and tryphine components such as sporopollenin, fatty acid and long-chain fatty acid derivatives are from the degradation of the sporophytic tapetum cells during pollen development [1]. Therefore, the exine development could be affected by the degeneration of tapetum cells in a sporophytic manner as in the mutation of GAMYB [7], which affects tapetum degradation and results in no pollen grain development. However, the thinner exine in the "-19" pollen (Fig 4A and 4B) can be attributed to the defective expression of the mutated *osga3ox1* gene in the "-19" pollen, but not in the "-3" pollen, by a gametophytic manner (Fig 3A, the "-3/-19" variant). The phenomena of the partially affected thinner exine (Fig 4A and 4B) and the various degrees of starch granule reduction in the "-19" pollen (Fig 3B and 3D) imply that the GA biosynthesis was significantly downregulated by the mutation of *osga3ox1*, but not completely absent, owing to the residual minor expression of other GA biosynthesis genes [14, 16, 20], such as *OsGA3ox2*. Moreover, the thickness of exine of almond pollen can be enhanced by $GA_3$-treatment [34]. These lines of evidence support that the exine formation is modulated by the availability of GA and also GA regulation, and the thinner exine in "-19" pollen was caused by the mutation of *OsGA3ox1* (Fig 4A and 4B), which reduced the amount of bioactive GA in the "-19" pollen at the gametophytic phase.

In term of starch synthesis, a recent study proposed that the cell wall invertase OsCIN3, which was not properly retained in the cell wall, might affect pollen development in *osga3ox1* [21]. OsINV4 (same as OsCIN3, LOC_Os04G33720) was expressed in mature pollen grains, and downregulation of OsINV4 disrupted starch formation in pollen [35]. We therefore propose that improper retention of OsCIN3/OsINV4 in *osga3ox1* might be due to structural changes in the exine of pollen walls, which caused impaired starch synthesis and finally led to defective accumulation of starch granules in *osga3ox1* pollen.

## Expression of *OsGAMYB* is downregulated in anther of the *osga3ox1* mutant

To explore the molecular regulation of pollen wall development, the expression of pollen development-related TF genes, such as *OsGAMYB*, *OsTDR* and *OsbHLH142*, in anthers of the *osga3ox1* mutant was investigated. These TF-encoding genes are directly or indirectly regulated by GA and are crucial for pollen wall development [7, 12, 13]. GAMYB, a GA-regulated TF [36], is a positive regulator of *OsTDR* and *OsbHLH142* [7, 11] and has been reported to regulate exine formation via activation of *CYP703A3* [7]. In addition, the lipid transporter proteins OsC4 and OsC6 are important for the development of pollen exine [37, 38], and downregulation of *OsC4* and *OsC6* was reported in mutants of *OsGAMYB*, *OsTDR* and *OsbHLH142* [7, 13, 39], suggesting that *OsTDR* and *OsbHLH142* might also be involved in exine development. Our qPCR data showed that the expression levels of *OsGAMYB*, *OsTDR* and *OsbHLH142* in *osga3ox1* were significantly downregulated (Fig 4D and S6 Fig), which might result in the thinner exine in *osga3ox1*.

GAMYB is a pivotal TF involved in the regulation of almost all GA-regulated genes during pollen development and it functions mainly in tapetum [7]. However, the expression of *GAMYB* and its target gene *CYP703A3* were also observed in microspores [7] suggesting that GAMYB and CYP703A3 are also involved in pollen development at the gametophytic phase. Given that GA regulates anther development and exine formation *via* GAMYB signaling [7],

the reduction of GA biosynthesis in pollen of *osga3ox1* mutant would downregulate the expression of GAMYB and possibly many of its downstream genes, such as *OsTDR*, *OsbHLH142*, *OsCYP703A3*, *OsC4* and *OsC6*, resulting in abnormal exines in the mutant.

## Conclusion

The mutation of *OsGA3ox1* affected exine development, which might affect the proper retention of invertase OsCIN3/OsINV4 on the cell wall, further affecting starch granule formation and resulting in low pollen viability and the seed-setting rate. These observations suggest that *OsGA3ox1* is a crucial gene for pollen development, although it does not affect the development of most vegetative and floral organs.

## Supporting information

**S1 Fig. Schematic diagrams of the *OsGA3ox1* gene structures, sgRNA target sites and the template vector pRGEB32 and analysis of the Cas9-induced *osga3ox1* variants.** A) The gene structure of *OsGA3ox1* showing exons (boxes), introns (lines), untranslated regions (UTRs) and sequences of sgRNA (red letters are the PAMs) at their target site (red vertical line). The primers (3ox1-CF and 3ox1-CR) used for DNA amplification and the expected sizes (in bp) of the PCR products are shown. A total of 1964 bp is used to draw this diagram. B) The template vector pRGEB32 for CRISPR/Cas9 constructs. Cas9 driven by the ubiquitin promoter ($P_{ubi}$), gRNA (red) and scaffold (blue) driven by the U3 promoter ($P_{OsU3}$) and hygromycin phosphotransferase (hpt) driven by the caMV35S promoter are indicated. The primers (arrows) used for detecting the T-DNA construct (F and R) and *hpt* gene (hpt-F and hpt-R) are shown. C) Characterization of 8 Cas9-induced transgenic T0 lines. PCR products were amplified from their construct and target gene regions using the primers indicated in B) and A), respectively. After PCR product sequencing, each of their sequencing results, either insertion (In) "+" or deletion (Del) "-", is indicated below each line using Arabic numerals separated by a slash (±n/±n) to represent their various diploid allelic genotypes. The superscript star sign (*) indicates that additional unidentified sequence modifications may exist in the line. The T1 seed-setting rate (SSR) for each line was measured and is provided below. D) The detailed In/Del sequences around the target site for each Cas9-induced line. Sequence data were obtained by Sanger sequencing using either the PCR products directly and/or plasmid DNA with cloned PCR products through the TA vector. E) Derived amino acid sequences based on the DNA sequences identified for various allelic templates. Deletions of amino acids are indicated with dashed lines. The "-3" deletions were all the same among different lines. The conserved amino acid sequences for all alleles at the N-terminal are indicated with red print, the replaced amino acid sequences due to frameshift are indicated with blue print, and star signs (*) indicate the stop codon. The lengths of the translated amino acids for each allele are provided. Locations of the two conserved domains DIOX_N and 2OG-FeII_Oxy in WT are indicated. (PPTX)

**S2 Fig. Analysis of T1 progenies from the Cas9-induced biallelic "-3/-19" variant (from line #1 T0 plant).** A) PCR product analysis of 8 transgenic T1 progenies for their constructs and target genes. Plants were segregated into "-3/-3", "-19/-19" and "-3/-19" recognized by gel patterns and confirmed by sequencing. The sequence In/Del for each variant is indicated. B) Representative T1 plants from different genotypes with their plant height and seed-setting rate (SSR). Plants were grown in the greenhouse. (PPTX)

**S3 Fig. Analysis of T2 progenies from heterozygotic/biallelic "-3/-19" T1 plants.** A) PCR product analysis of 24 T2 progenies from the "3/-19" biallelic variant (T1 plant #2 in S2 Fig). T2 Plants were segregated to 11:11:2 = "-3/-3":"-3/-19":"-19/-19" based on the gel patterns and confirmed by sequencing. The sequence In/Del for each plant is indicated below. B) Representative T2 plants from different genotypes "-3/-3": "-3/-19": "-19/-19". Plants were grown in the paddy field. Bar = 20 cm.
(PPTX)

**S4 Fig. Comparison of pollen grains in microscopic anther sections from WT and *osga3ox1* variants of different genotypes.** The anther microscopic sections of WT, heterozygous ("-3/-19" and "-3/-10") and homozygous ("-3/-3", "-19/-19" and "-2/-2") *osga3ox1* variants are shown. Two types of pollen grains, the normal pollen grain ("-3"-like, blue arrows) and abnormal/starch-less pollen grain ("-10"-like and "-19"-like, red arrows) are observed within pollen sacs of "-3/-19" and "-3/-10" heterozygous mutants. The "-3"-like and "-19"-like pollens can be visually identified, examples of different pollen type from each sac are indicated with corresponding base deletion symbols. A) one anther sac with better resolution, Bar = 20 μm. B) sections with 4 anther sacs, Bar = 70 μm.
(PPTX)

**S5 Fig. Comparison of the ultrastructure of pollen wall of WT, -3, -19, "-3"-like and "-19"-like pollens.** The "-3" and "-19" pollens were from homozygous "-3/-3" and "-19/-19" anther, respectively. The "-3"-like and "-19"-like pollens were from the heterozygous "-3/-19" mutant. The structure of exine, including tectum (Te), tryphine (Ty), baculum (Ba) and nexine (Ne), are indicated in the WT pollen. The abnormal formation of baculum (blue arrow) and tryphine (red arrow) in "-19" and "-19"-like pollen are pointed. Bar = 500 nm.
(PPTX)

**S6 Fig. Expression analysis of pollen development genes in *osga3ox1* mutants.** Expression analysis of the *OsGAMYB*, *OsTDR*, and *OsbHLH142* pollen development genes in WT and the "-10/-10" and "-19/-19" *ga3ox1* knockout mutants. The significant differences between WT and knockout mutants were determined by Student's *t test*. $^*p<0.05$, $^{**}p<0.01$, $^{***}p<0.001$.
(PPTX)

**S1 Table. List of primers and their sequences used in this study.**
(PPTX)

**S1 Raw images.**
(PPTX)

## Acknowledgments

We thank the Division of Biotechnology, TARI, for providing the greenhouse and isolated field to grow the transgenic rice plants.

## Author Contributions

**Conceptualization:** Kun-Ting Hsieh, Liang-Jwu Chen.

**Data curation:** Kun-Ting Hsieh, Chi-Chih Wu, Shih-Jie Lee, Yu-Heng Chen, Shiau-Yu Shiue, Yi-Chun Liao, Su-Hui Liu, I.-Wen Wang, Liang-Jwu Chen.

**Formal analysis:** Kun-Ting Hsieh, Shih-Jie Lee, Yu-Heng Chen, Shiau-Yu Shiue.

**Funding acquisition:** Wen-Hsiung Li, Chang-Sheng Wang, Liang-Jwu Chen.

**Investigation:** Kun-Ting Hsieh, Wen-Hsiung Li, Chang-Sheng Wang, Liang-Jwu Chen.

**Methodology:** Kun-Ting Hsieh, Chi-Chih Wu, Shih-Jie Lee, Yu-Heng Chen, Shiau-Yu Shiue, Yi-Chun Liao, Su-Hui Liu, Liang-Jwu Chen.

**Project administration:** Chang-Sheng Wang, Liang-Jwu Chen.

**Resources:** I.-Wen Wang, Ching-Shan Tseng, Wen-Hsiung Li, Chang-Sheng Wang, Liang-Jwu Chen.

**Supervision:** Ching-Shan Tseng, Liang-Jwu Chen.

**Writing – original draft:** Kun-Ting Hsieh.

**Writing – review & editing:** Kun-Ting Hsieh, Chi-Chih Wu, Wen-Hsiung Li, Chang-Sheng Wang, Liang-Jwu Chen.

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
