## [Decision Letter · Decision Letter 0]

3 Sep 2023

PONE-D-23-25784Rice GA3ox1 modulates pollen starch granule accumulation and pollen wall developmentPLOS ONE

Dear Dr. Chen,

Thank you for submitting your manuscript to PLOS ONE. After careful consideration, we feel that it has merit but does not fully meet PLOS ONE’s publication criteria as it currently stands. Therefore, we invite you to submit a revised version of the manuscript that addresses the points raised during the review process.

 the manuscript that addresses the points raised during the review process.

The results of the manuscript are intriguing; however, the reviewer has raised a few important points that must be clarified. Therefore, this manuscript required revision prior to publication consideration by PLOS ONE.  Please refer to the attached report for detail.

We look forward to receiving your revised manuscript.

Kind regards,

Muhammad Qasim Shahid

Academic Editor

PLOS ONE

Journal Requirements:

"The authors declare that they have no conflict of interest."

4. Please remove your figures from within your manuscript file, leaving only the individual TIFF/EPS image files, uploaded separately. These will be automatically included in the reviewers’ PDF.

5. We notice that your supplementary [Fig. S1-S5] are included in the manuscript file. Please remove them and upload them with the file type 'Supporting Information'. Please ensure that each Supporting Information file has a legend listed in the manuscript after the references list.

Additional Editor Comments:

The results of the manuscript are intriguing; however, the reviewer has raised a few important points that must be clarified. Therefore, this manuscript required revision prior to publication consideration by PLOS ONE. Please refer to the attached report for detail.

Reviewers' comments:

Reviewer's Responses to Questions

**Comments to the Author**

1. Is the manuscript technically sound, and do the data support the conclusions?

Reviewer #1: Yes

Reviewer #2: Yes

2. Has the statistical analysis been performed appropriately and rigorously? 

Reviewer #1: Yes

Reviewer #2: Yes

3. Have the authors made all data underlying the findings in their manuscript fully available?

Reviewer #1: Yes

Reviewer #2: Yes

4. Is the manuscript presented in an intelligible fashion and written in standard English?

Reviewer #1: Yes

Reviewer #2: Yes

5. Review Comments to the Author

Reviewer #1: -The authors analyzed the molecular functions of rice GA3ox1, essential for the development of tapetum and exine as well as the starch accumulation. osga3ox1 mutants generated by CRISPR/Cas9 showed a reduced number of total pollen grains and a significantly lower seed setting rate. Furthermore, the histochemical analysis revealed that the osga3ox1 mutant induces tapetum deficient anther and that the pollen grains with osga3ox1 allele showed morphological changes in the exine layer including bacula and tryphines, which can cause disrupted pollen wall development. And the transcriptional reductions of OsGAMYB, OsTDR, and OsbHLH142 support morphological defects of osga3ox1 knockout anther and pollen grain.

-Previously, one article showed that the same GA3ox1 has a higher GA7 synthesis ratio and the bioactive GA synthesis by GA3ox1 is critical for starch accumulation in rice pollen (Kawai, K., Takehara, S., Kashio, T. et al. Evolutionary alterations in gene expression and enzymatic activities of gibberellin 3-oxidase 1 in Oryza. Commun Biol 5, 67 (2022). https://doi.org/10.1038/s42003-022-03008-5). Current manuscript added cellular gametophytic features in GA3ox1 mutants, which further confirmed the previous finding. I would like to ask authors to improve few things.

-L34: While the SSR of the knockout mutants were dramatically decreased (Fig. 1H), a high proportion (~51.3%) of the stained pollen were observed in the present study (Fig. 2D) which differs from the low proportion (~5%) reported previously [21]. This variation could be due to the different rice cultivars (TNG71 vs Nipponbare) and culture environments (Taichung Taiwan 24°0′N 120°7′E vs Nagoya Japan 35°2′N 136°9′E).

Though this manuscript suggested that the phenotypical difference from Kawai et al. (2022) might be from the different background cultivars or growth environment, I think the differences may come from the unique features of the CRISPR/Cas9 system. Many publications reporting the CRISPR/Cas9-mediated mutants have used transgene-free homozygous/biallelic (or heterozygous) T1 (or T2) plants to prevent functional variations due to chimeric mutations from the constitutive expressions of sgRNA and Cas9 of the remaining T-DNA. In this case, a transgene-free homozygote (Figure 1B #3-5) and transgene-free heterozygotes (Figure 1A #1-2, 1B #3-3 and #3-8) are proper materials for further investigations. However, it seems that the authors used T1 plants containing the transgene, according to Figure S2A and Figure 1A-C, unlike the former publication used transgene-free T1 lines as their materials. Thus, this data needs to be re-verified using such transgene-free homozygous mutants lines.

-Fig 4D: For RT-qPCR, currently it has only one WT and one mutant. To clarify this result, repeat this experiment from at least two homozygous mutants.

Reviewer #2: This paper described a detailed study on the mutants of OsGA3ox1 (generated by the authors using the CRISPR/Cas9 technique) with aim to answer the following questions: Whether OsGA3ox1 knockout would affect plant growth and grain setting? How the knockout affects pollen development? What is the molecular mechanism underpinning pollen sterility? The study is well designed and implemented using state of the art techniques, the findings were reported against sound analysis of background studies and were well supported by the data, and the writing is easy for understanding. Data are properly analyzed using statistic means and are nicely presented in various formats.

6. PLOS authors have the option to publish the peer review history of their article (what does this mean?). If published, this will include your full peer review and any attached files.

Reviewer #1: No

Reviewer #2: **Yes: **Qing-Yao Shu

---

## [Author Response · Author response to Decision Letter 0]

17 Sep 2023

Response to Reviewers’ comments:

Reviewers’ Comments

Reviewer #1: 

- The authors analyzed the molecular functions of rice GA3ox1, essential for the development of tapetum and exine as well as the starch accumulation. osga3ox1 mutants generated by CRISPR/Cas9 showed a reduced number of total pollen grains and a significantly lower seed setting rate. Furthermore, the histochemical analysis revealed that the osga3ox1 mutant induces tapetum deficient anther and that the pollen grains with osga3ox1 allele showed morphological changes in the exine layer including bacula and tryphines, which can cause disrupted pollen wall development. And the transcriptional reductions of OsGAMYB, OsTDR, and OsbHLH142 support morphological defects of osga3ox1 knockout anther and pollen grain.

- Previously, one article showed that the same GA3ox1 has a higher GA7 synthesis ratio and the bioactive GA synthesis by GA3ox1 is critical for starch accumulation in rice pollen (Kawai, K., Takehara, S., Kashio, T. et al. Evolutionary alterations in gene expression and enzymatic activities of gibberellin 3-oxidase 1 in Oryza. Commun Biol 5, 67 (2022). https://doi.org/10.1038/s42003-022-03008-5). Current manuscript added cellular gametophytic features in GA3ox1 mutants, which further confirmed the previous finding. 

I would like to ask authors to improve few things.

-L34: While the SSR of the knockout mutants were dramatically decreased (Fig. 1H), a high proportion (~51.3%) of the stained pollen were observed in the present study (Fig. 2D) which differs from the low proportion (~5%) reported previously [21]. This variation could be due to the different rice cultivars (TNG71 vs Nipponbare) and culture environments (Taichung Taiwan 24°0′N 120°7′E vs Nagoya Japan 35°2′N 136°9′E).

Question #1:

Though this manuscript suggested that the phenotypical difference from Kawai et al. (2022) might be from the different background cultivars or growth environment, I think the differences may come from the unique features of the CRISPR/Cas9 system. Many publications reporting the CRISPR/Cas9-mediated mutants have used transgene-free homozygous/biallelic (or heterozygous) T1 (or T2) plants to prevent functional variations due to chimeric mutations from the constitutive expressions of sgRNA and Cas9 of the remaining T-DNA. In this case, a transgene-free homozygote (Figure 1B #3-5) and transgene-free heterozygotes (Figure 1A #1-2, 1B #3-3 and #3-8) are proper materials for further investigations. However, it seems that the authors used T1 plants containing the transgene, according to Figure S2A and Figure 1A-C, unlike the former publication used transgene-free T1 lines as their materials. Thus, this data needs to be re-verified using such transgene-free homozygous mutants lines.

Response to question #1:

Thank you for the comments. We may not have provided a sufficiently clear description of the materials we analyzed. Usually, the homozygous lines produce very few seeds or are sterile, as such they are difficult to be used for progenies analysis. Therefore, the materials used for most of our experiments and progenies analysis were from heterozygous parental lines. For example, the “-3/-19” biallelic variant presented in Figure 1A was T2 progenies segregated from transgenic T1 heterozygous lines #1-1, #1-2 and #1-5 (Fig S2A). Then the transgene-free T2 heterozygote, such as the line #1-2 from Figure 1A, was selected to obtain various genotypes of progenies and their phenotypical data and SSR were presented in the Figures 1D and 1H respectively. The other two biallelic variants “-3/-2” and “-3/-10” followed the same processes to get the various genotype of T2 progenies and data were presented in Figures 1E, 1F and 1I. 

So, the “-3/-19” materials used in the present study were all transgene-free hetero/homozygotes (Figs 2 to 4 and Table 1), as well as the “-3/-2” variant but not the “-3/-10” variant (the transgene-free “-3/-10” lines were not yet available).

The materials used for Figure 2D were transgene-free hetero/homozygotes. The percentage of stained pollens differ from previously reported (Kawai et al. 2022) could be due to the different rice cultivars and culture environments. It has been reported that different genetic background (Kawai et al. 2022) and colder temperature (Sakata et al., 2014) affect GA biosynthesis and these differences might also affect the SSR, leading to of ~80% in TNG71 (Fig. 1H) vs ~60% in Nipponbare (Kawai et al., 2022). Furthermore, besides a few heavy stained pollen, most pollens in the knockout mutant were stained lighter than WT (Fig. 2B) and these pollens were counted as stained pollen even though they might not be viable.

To clarify the description in the manuscript, we added the “transgene-free” wording in the sentence of “Detailed agronomic traits of the WT host and progenies segregated from the transgene-free “-3/-19” variant/line were compared, and the plant height, panicle number and grain number/per panicle were the same for WT and all mutant genotypes (Table 1).” in line 10, page 9; and in the sentence of “To determine whether the low SSR was due to defects in any gametophytic features, the flower organs of the three genotypes segregated from the progeny of the transgene-free “-3/-19” variant were examined.” in line 22, page 9.

Question #2:

-Fig 4D: For RT-qPCR, currently it has only one WT and one mutant. To clarify this result, repeat this experiment from at least two homozygous mutants.

Response to question #2:

Thank you for the comment. We have compared the expression of the three TFs (OsGAMYB, OsTDR, and OsbHLH142) in WT, “-10/-10” and “-19/-19”, and both “-10/-10” and “-19/-19” showed similar results. We presented only the data of “-19/-19” in the manuscript, since “10/-10” was not yet a transgene-free mutant. We have added these results as supplementary information (Fig. S6) and also indicated in the text of line 18, page 13.

---

## [Decision Letter · Decision Letter 1]

20 Sep 2023

Rice GA3ox1 modulates pollen starch granule accumulation and pollen wall development

PONE-D-23-25784R1

Dear Dr. Chen,

We’re pleased to inform you that your manuscript has been judged scientifically suitable for publication and will be formally accepted for publication once it meets all outstanding technical requirements.

Kind regards,

Muhammad Qasim Shahid

Academic Editor

PLOS ONE

Additional Editor Comments (optional):

Authors have done the suggested changes, so MS could be accepted for publication.

Reviewers' comments:

Reviewer's Responses to Questions

**Comments to the Author**

1. If the authors have adequately addressed your comments raised in a previous round of review and you feel that this manuscript is now acceptable for publication, you may indicate that here to bypass the “Comments to the Author” section, enter your conflict of interest statement in the “Confidential to Editor” section, and submit your "Accept" recommendation.

Reviewer #1: All comments have been addressed

2. Is the manuscript technically sound, and do the data support the conclusions?

Reviewer #1: Yes

3. Has the statistical analysis been performed appropriately and rigorously? 

Reviewer #1: N/A

4. Have the authors made all data underlying the findings in their manuscript fully available?

Reviewer #1: Yes

5. Is the manuscript presented in an intelligible fashion and written in standard English?

Reviewer #1: Yes

6. Review Comments to the Author

Reviewer #1: Authors well addressed all comments I raised previously. I think this revised manuscript is now ready for publication.

7. PLOS authors have the option to publish the peer review history of their article (what does this mean?). If published, this will include your full peer review and any attached files.

Reviewer #1: No
